# Balancing Model Efficiency and Performance: Adaptive Pruner for Long-tailed Data

**Zhe Zhao** [* 1 2]  **HaiBin Wen** [* 2]  **Pengkun Wang** [1 3]  **Shuang Wang** [1]  **Zhenkun Wang** [4]  **Qingfu Zhang** [2]
**Yang Wang** [1 3 5]

## Abstract

Long-tailed distribution datasets are prevalent in many machine learning tasks, yet existing neural network models still face significant challenges when handling such data. This paper proposes a novel adaptive pruning strategy, LTAP (Long-Tailed Adaptive Pruner), aimed at balancing model efficiency and performance to better address the challenges posed by long-tailed data distributions. LTAP introduces multi-dimensional importance scoring criteria and designs a dynamic weight adjustment mechanism to adaptively determine the pruning priority of parameters for different classes. By focusing on protecting parameters critical for tail classes, LTAP significantly enhances computational efficiency while maintaining model performance. This method combines the strengths of long-tailed learning and neural network pruning, overcoming the limitations of existing approaches in handling imbalanced data. Extensive experiments demonstrate that LTAP outperforms existing methods on various long-tailed datasets, achieving a good balance between model compression rate, computational efficiency, and classification accuracy. This research provides new insights into solving model optimization problems in long-tailed learning and is significant for improving the performance of neural networks on imbalanced datasets. The code is available at https://github.com/DataLab-atom/LT-VOTE.

---

[*]Equal contribution  [1]University of Science and Technology of China, Hefei, China [2]City University of Hong Kong, Hong Kong [3]Suzhou Institute for Advanced Research, USTC, Suzhou, China [4]Southern University of Science and Technology, Shenzhen, China [5]Anhui Provincial Key Laboratory of High Performance Computing, Hefei, China. Correspondence to: Pengkun Wang <pengkun@ustc.edu.cn>, Yang Wang <angyan@ustc.edu.cn>.

*Proceedings of the 42^{nd} International Conference on Machine Learning*, Vancouver, Canada. PMLR 267, 2025. Copyright 2025 by the author(s).

## 1. Introduction

Long-tailed learning aims to address the problem of highly imbalanced class distributions, where most classes (i.e., tail classes) have scarce samples, while few classes (i.e., head classes) have abundant samples (Liu et al., 2019; Wang et al., 2017; Tan et al., 2020; Li et al., 2020). This data distribution is prevalent in real-world applications, such as e-commerce product classification, speech recognition, and natural language processing (Ouyang et al., 2016; Yang & Xu, 2020). Although deep learning models perform excellently on head classes, their performance on tail classes remains limited, mainly because models tend to overfit head classes and neglect feature learning for tail classes, leading to insufficient overall model generalization (Kang et al., 2019).

To improve long-tailed learning performance, researchers have proposed various methods, including multi-expert systems and modular designs (Wang et al., 2020; Liu et al., 2019). However, these traditional methods face numerous challenges in practical applications. For instance, multi-expert systems often require training and maintaining multiple independent sub-models, resulting in enormous computational and storage resource consumption (Xiang et al., 2020). Modular designs rely on predefined module structures, lacking dynamic adaptability and struggling to cope with constantly changing data distributions (Ren et al., 2020). Moreover, *these methods often struggle to efficiently utilize parameters when dealing with tail classes, limiting model performance on scarce data (Zhang et al., 2021a).*

As a model compression and optimization technique, pruning optimizes model structure and improves computational efficiency by removing redundant or unimportant neurons or connections (Han et al., 2015; Liu et al., 2018; Zhu & Gupta, 2017). In recent years, pruning methods have shown significant advantages in improving model performance, reducing parameter counts, and accelerating inference (Frankle & Carbin, 2018; Blalock et al., 2020). However, ordinary pruning methods face special **challenges** when applied to long-tailed learning:

- *Pruning bias due to class imbalance*: conventional pruning methods, blind to class-specific contributions,

risk exacerbating the very imbalance they aim to address by inadvertently removing neurons crucial for tail class recognition (He et al., 2021a).

- *Difficulty in dynamic adjustment*: the static nature of traditional pruning methods conflicts with the dynamic evolution of the association between parameters and data distribution during training, potentially leading to suboptimal or even harmful pruning decisions (Molchanov et al., 2019).

- *Single evaluation criterion*: the reliance on simplistic pruning criteria fails to capture the nuanced importance of neurons in the complex landscape of long-tail distributions, potentially misleading the pruning process (Frankle et al., 2020).

To address these challenges, this paper proposes a novel pruning strategy called **L**ong-**T**ailed **A**daptive **P**runer (**LTAP**), specifically optimized for long-tailed learning environments. LTAP is rooted in the understanding that effective long-tailed learning requires a fundamental rethinking of how we allocate and utilize model capacity (Kang et al., 2019). Our method makes **innovative contributions** in the following aspects:

- *New LT-Vote mechanism*: Through the **LT-Vote** (Long-Tailed Voting) mechanism, we dynamically adjust the weights of different pruning criteria based on the classification accuracy of different classes, enabling the pruning process to more specifically optimize the learning performance of tail classes, enhancing model robustness on long-tail distribution data.

- *Multi-stage dynamic pruning*: Our method divides the pruning process into multiple stages, gradually removing redundant parameters, and dynamically adjusts the pruning strategy at each stage based on current model performance, ensuring continuous performance optimization during the pruning process.

- *Efficient resource utilization*: By reducing model parameter count and computational requirements through pruning, we improve model operational efficiency in resource-constrained environments while maintaining or even improving classification accuracy on tail classes.

Experimental results show that our proposed LTAP method significantly outperforms traditional pruning methods and other long-tailed learning methods on multiple long-tailed distribution datasets, validating its effectiveness in enhancing tail class recognition ability, optimizing model structure, and improving computational efficiency.

## 2. LTAP: Adaptive Pruner for Long-tailed Distribution

We propose LTAP, a novel pruning framework that adaptively protects crucial parameters while achieving efficient model compression. In this section, we first formalize the problem setup, then present our multi-criteria importance evaluation framework, followed by our dynamic weight adjustment mechanism and progressive pruning strategy.

### 2.1. Problem Formulation

Consider a deep neural network $f_\theta : \mathcal{X} \to \mathcal{Y}$ parameterized by $\theta$, trained on a long-tailed dataset $\mathcal{D} = \{(x_i, y_i)\}_{i=1}^N$. The number of samples per class in $\mathcal{D}$ follows a power law distribution, leading to significant imbalance between head and tail classes. The network parameters are organized into groups $\mathcal{G} = \{g_1, ..., g_M\}$, where each group may correspond to a layer, filter, or other structural components.

Our objective is to identify a binary mask $\mathbf{m} \in \{0, 1\}^{|\theta|}$ that: 1. Achieves a target sparsity ratio $\gamma_{\text{total}}$ 2. Maintains or improves model performance, especially on tail classes 3. Preserves essential parameter interactions across different network components

This presents a constrained optimization problem:

$$\min_{\mathbf{m}} \quad \mathcal{L}(f_{\theta \odot (1-\mathbf{m})}; \mathcal{D})$$
$$\text{s.t.} \quad \|\mathbf{m}\|_0 \leq \gamma_{\text{total}} |\theta| \quad (1)$$
$$\mathcal{A}_c(f_{\theta \odot (1-\mathbf{m})}) \geq \tau_c, \forall c \in \mathcal{C}$$

where $\mathcal{A}_c$ represents the accuracy on class $c$, and $\tau_c$ is a class-specific performance threshold.

### 2.2. Multi-criteria Importance Evaluation

Traditional pruning methods often rely on singular criteria such as weight magnitude or gradient-based importance. However, in long-tailed scenarios, different criteria may be more relevant for different classes or network regions. We propose a comprehensive importance scoring framework that integrates multiple complementary perspectives:

**Definition 1 (Parameter Group Score).** For each parameter group $g \in \mathcal{G}$, we define its importance score as a weighted combination of multiple criteria:

$$S_g = \sum_{k=1}^K \alpha_k \cdot s_{g,k}, \quad \alpha_k \in \mathbb{R}_+, \sum_{k=1}^K \alpha_k = 1 \quad (2)$$

The scoring criteria can be categorized into two main families:

1. **Magnitude-based Criteria:** These capture the static

importance of parameters:

$$s_{g,\text{mag}} = \|\mathbf{w}_g\|_2 \in \mathbb{R}_+ \quad \text{(absolute magnitude)}$$

$$s_{g,\text{avg-mag}} = \frac{\|\mathbf{w}_g\|_2}{n_g} \in \mathbb{R}_+ \quad \text{(normalized magnitude)} \quad (3)$$

**2. Gradient-based Criteria:** These reflect the dynamic importance during training:

$$s_{g,\cos} = \frac{\mathbf{w}_g \cdot \nabla_{\mathbf{w}_g}\mathcal{L}}{\|\mathbf{w}_g\|_2 \|\nabla_{\mathbf{w}_g}\mathcal{L}\|_2} \quad \text{(update alignment)}$$

$$s_{g,\text{taylor-1}} = |\nabla_{\mathbf{w}_g}\mathcal{L}| \cdot |\mathbf{w}_g| \quad \text{(first-order impact)}$$

$$s_{g,\text{taylor-2}} = |\nabla^2_{\mathbf{w}_g}\mathcal{L}| \cdot \mathbf{w}_g^2 \quad \text{(second-order stability)}$$

$$(4)$$

Each criterion provides unique insights: magnitude-based criteria identify dominant parameters, while gradient-based criteria capture training dynamics and optimization geometry. The cosine similarity term $s_{g,\cos}$ specifically measures how well parameter updates align with the current optimization trajectory, helping identify parameters that consistently contribute to learning.

### 2.3. Dynamic Weight Adjustment Mechanism

A key innovation in LTAP is its adaptive weighting scheme that dynamically adjusts the importance of different criteria based on class-specific performance. This ensures that parameters crucial for tail classes are properly evaluated and protected.

**Definition 2 (Class-Criteria Interaction).** We model the interaction between classes and criteria through a matrix multiplication:

$$I_c = DN_c, \quad D \in \mathbb{R}^{K \times C}, N_c \in \mathbb{R}^C$$
$$\alpha^{(t)} = \text{softmax}(I_c) \in \mathbb{R}^K \quad (5)$$

Here, $D$ represents a learnable criteria weight matrix that captures the effectiveness of each criterion for each class, and $N_c$ is the class distribution vector that encodes the long-tailed nature of the dataset.

The weights are updated through a performance-driven mechanism:

$$D_k^{(t+1)}[c] = D_k^{(t)}[c] + \beta \cdot \mathbb{K}(A_c^{(t)} > A_c^{(t-1)}) \quad (6)$$

This update rule has several important properties: 1. It strengthens criteria that lead to improved class performance 2. It maintains separate criterion preferences for different classes 3. It naturally adapts to the learning dynamics throughout training

### 2.4. Progressive Multi-stage Pruning Strategy

To ensure stable model compression while protecting tail-class performance, LTAP employs a progressive pruning strategy that distributes the overall pruning ratio $\gamma_{\text{total}}$ across $P$ stages:

$$\gamma_p = \frac{\gamma_{\text{total}}}{P} \quad (7)$$

where $\gamma_p$ denotes the pruning ratio for each stage. The pruning process consists of four key steps:

1. Compute importance scores $S_g$ for each parameter group $g \in \mathcal{G}$ based on current model state

2. Select redundant parameter groups $\mathcal{R} = \{g \mid S_g \leq S_{\text{threshold}}, |\mathcal{R}|/|\mathcal{G}| \approx \gamma_p\}$

3. Zero out weights of redundant groups: $\theta_g \leftarrow \theta_g \odot (1 - \mathbf{m}_g)$ for $g \in \mathcal{R}$

4. Update remaining parameters according to the pruned structure

The optimization process alternates between training and pruning:

$$\theta^{(t+1)} = \theta^{(t)} - \eta \nabla_\theta \mathcal{L}(\theta^{(t)}) \quad (8)$$

where $\theta^{(t)}$ represents model parameters at step $t$, $\eta$ is the learning rate, and $\mathcal{L}$ is the loss function. During parameter updates, LTAP incorporates both gradient information and pruning masks:

$$\nabla_g S_g = \sum_{k=1}^K \alpha_k \cdot \nabla_g s_{g,k}$$
$$\theta_g \leftarrow \theta_g - \eta \cdot \nabla_g S_g \quad (9)$$
$$\theta_g \leftarrow \theta_g \odot (1 - \mathbf{m}_g)$$

where $\nabla_g S_g$ combines $K$ different importance criteria weighted by $\alpha_k$, and $\mathbf{m}_g \in \{0,1\}^{d_g}$ is the pruning mask. The LT-Vote mechanism dynamically adjusts $\{\alpha_k\}_{k=1}^K$ based on validation performance to protect tail-class parameters.

## 3. Theoretical Analysis: Tail Classes Benefit More from Overparameterization

To verify the effectiveness and soundness of our method, we first establish a series of foundational definitions. Then, through lemmas and theorems, we systematically argue that tail classes in long-tailed distributions have higher requirements for model overparameterization. Based on this, we

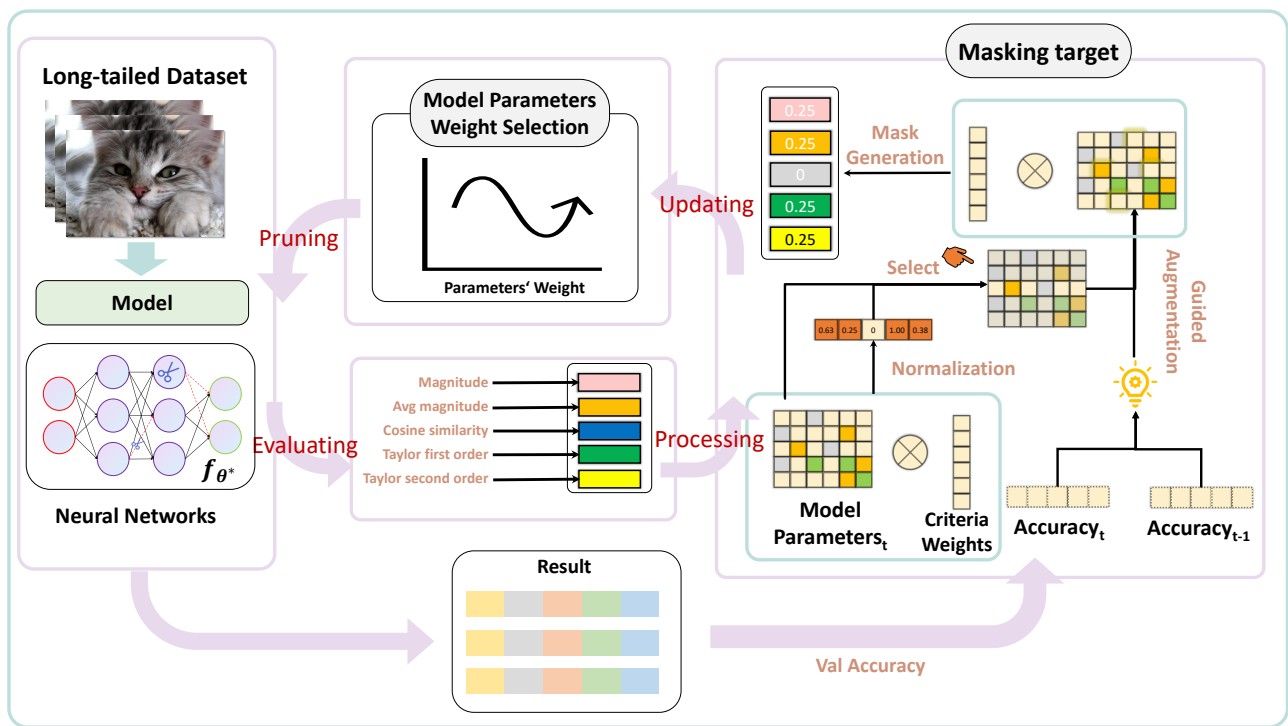

*Figure 1.* Overview of the Long-Tailed Adaptive Pruner (LTAP) methodology. The diagram illustrates the iterative process of model evaluation, parameter processing, and pruning, highlighting the integration of multiple importance criteria and the dynamic weight adjustment mechanism.

propose a differentiated parameter allocation strategy and the tail-biased pruning proposition. Finally, we synthesize these theoretical results and prove the effectiveness of the tail-biased pruning strategy in learning from long-tailed distributions.

To gain a deeper understanding of the learning difficulty of different classes under long-tailed distributions, we introduce the sample complexity lemma (Lemma 1 in Appendix), establishing the relationship between sample size, VC dimension, and generalization error. Based on this lemma, we further define the class-specific VC dimension (Definition 1 in Appendix) and derive the learning difficulty for each class (Corollary 1 in Appendix).

**Theorem 1** (Differentiated Overparameterization Demand). *In a long-tailed setting, to achieve the same generalization performance, tail classes require a higher degree of overparameterization than head classes. Specifically, for tail classes, $\gamma_c \geq \Omega\left(\frac{N_1}{N_c} \cdot \frac{1}{\log N_c}\right)$, and for head classes, $\gamma_c \sim O(1)$, where $N_1$ is the number of samples in the head class (the class with the most samples), and $N_c$ is the number of samples in class c.*

The differentiated overparameterization demand theorem reveals that tail classes in a long-tailed dataset have a higher

demand for model overparameterization, which is further supported by the imbalance in overparameterization demand (Corollary 2 in Appendix).

Based on the differentiated overparameterization demand theorem, we propose the parameter allocation strategy corollary, which guides how to reasonably allocate model parameters in long-tailed learning.

**Theorem 2** (Parameter Allocation Strategy). *In long-tailed learning, to optimize overall model performance, relatively more parameters should be allocated to tail classes. Specifically, for class c, the ideal parameter allocation ratio $\alpha_c$ should satisfy $\alpha_c \propto \frac{N_1}{N_c} \cdot \frac{1}{\log N_c}$, where $N_1$ is the number of samples in the head class (the class with the most samples), and $N_c$ is the number of samples in class c.*

**Theorem 3** (Performance Gains from Parameter Allocation). *Assume that model performance is logarithmically related to the number of effective parameters for each class, i.e., for class c, its performance $perf_c$ satisfies $perf_c \propto \log P_c$, where $P_c$ is the number of effective parameters for class c. Under the above parameter allocation strategy, compared to uniform allocation, the performance gain $\Delta$ is $\Delta \geq \Omega\left(\frac{1}{C} \sum_{c=1}^{C} \log\left(\frac{N_1}{N_c}\right)\right)$, where $C$ is the total number of classes.*

This theorem shows that by reasonably allocating parameters, we can significantly improve overall model performance on long-tailed datasets, especially improving the classification accuracy of tail classes.

Based on the above theoretical foundations, we propose the ***tail-biased pruning proposition, which guides how to prune models in long-tailed learning***.

**Proposition 1** (Tail-biased Pruning). *In long-tailed learning, to reduce the number of parameters while maintaining overall model performance, a pruning strategy that favors retaining parameters for tail classes should be adopted. Specifically, the optimization objective is* $\min_{\mathbf{m}} \sum_{c=1}^{C} w_c L_c(f_{\theta \odot \mathbf{m}}(\mathbf{x}), \mathbf{y}) + \lambda \|\mathbf{m}\|_0$, *where* $\mathbf{m} \in \{0,1\}^{|\theta|}$ *is a binary mask vector indicating whether a parameter is retained,* $w_c \propto \frac{N_1}{N_c} \cdot \frac{1}{\log N_c}$ *is the weight for class* $c$, $L_c$ *is the loss function for class* $c$, $f_{\theta \odot \mathbf{m}}$ *denotes the masked model,* $\lambda$ *is a hyperparameter controlling the pruning strength, and* $\|\mathbf{m}\|_0$ *is the* $L_0$ *norm of* $\mathbf{m}$.

This proposition, through a weighted loss function and parameter sparsity, guides how to prioritize the retention of tail class parameters during pruning, ensuring that model parameters are reduced while maintaining or improving overall performance on long-tailed datasets.

To ensure the effectiveness of the tail-biased pruning strategy in practical applications, we propose the following performance guarantee theorem.

**Theorem 4** (Performance Guarantee of Tail-biased Pruning). *Assume that the initial model achieves a training error of* $\epsilon$ *on each class. After applying the tail-biased pruning strategy, the expected generalization error* $\mathbb{E}[\hat{\epsilon}_c]$ *for class* $c$ *satisfies* $\mathbb{E}[\hat{\epsilon}_c] \leq \epsilon + O\left(\sqrt{\frac{\log(N_c/\delta)}{N_c}}\right)$, *where* $\delta$ *is a small constant (e.g., 0.05), representing the confidence level.*

This theorem shows that despite the pruning process, the generalization error for tail classes can still be effectively controlled, and the strategy ensures that overall performance does not significantly degrade, especially in terms of the performance of tail classes. The relevant proofs can be found in **Appendix A**.

## 4. Experiments

In this section, we evaluate the performance of our proposed method on multiple long-tailed datasets. Furthermore, we assess the computational efficiency of each method by comparing the ratio of floating-point operations (FLOPs) and the ratio of accuracy improvement.

### 4.1. Experimental Setup

**Datasets.** CIFAR-100-LT is a long-tailed version of CIFAR-100, containing 100 classes with two imbalance ratios (IR = 50, 100). ImageNet-LT is a long-tailed version of ImageNet, with 1,000 classes and natural long-tailed distribution. iNaturalist 2018 is a large-scale real-world dataset with 8,142 species categories and inherent long-tailed distribution.

**Implementation Details.** We use the knowledge generated from the long-tailed recognition task to guide the pruning of the backbone network. Specifically, for each parameter in the model, we calculate scores using 'magnitude', 'avg_magnitude', 'cosine_similarity', 'taylor_first_order', and 'taylor_second_order' during the gradient descent process. These scores are then weighted based on the cumulative change in accuracy for each class on the validation set. The weighted sum of the scores is used to determine whether to prune a parameter. We start the continuous pruning process after the 100th epoch, and the final model retains 30% of the original parameters. For the final evaluation phase, we use the same settings as DODA (Wang et al., 2024) for all baseline methods and our method. For the CIFAR-100-LT dataset, we follow the general experimental settings of (Cao et al., 2019) and use ResNet-32 (proposed by (He et al., 2016)) as the backbone network. The network is trained for 200 epochs using the GD optimizer with an initial learning rate of $10^{-4}$, momentum of 0.9, and weight decay of $2 \times 10^{-4}$. For ImageNet-LT and iNaturalist 2018 datasets, we use ResNet-50 as the backbone network, train the network for 100 epochs with an initial learning rate of 0.1, and reduce the learning rate by 0.1 at the 60th and 80th epochs. For all experiments, we set the value of the hyperparameter pau to 0.5.

**Baselines.** For fair comparison, all methods are evaluated under the same experimental conditions. We use three strong long-tailed baselines, e.g., Balanced Softmax (BS) (Ren et al., 2020), LDAM-DRW (Cao et al., 2019), DBLP (Zhou et al., 2024) and two SOTA pruning method, ATO(Wu et al., 2024), RReg(Stewart et al., 2023). Our proposed method is denoted as 'BS + LTAP', 'LDAM-DRW + LTAP', and 'DBLP + LTAP'. We report the classification accuracy of the head, medium, and tail classes, as well as the overall accuracy across all classes. Additionally, we compute the ratio of accuracy to FLOPs ($\frac{C}{F}$) as a key metric to evaluate both performance and computational efficiency.

### 4.2. Benchmark Results

**CIFAR-100-LT.** Table 1 presents the classification results for different methods on the CIFAR-100-LT dataset under two imbalance ratios (IR = 50 and 100). LTAP consistently achieves higher $\frac{C}{F}$ compared to other pruning methods in both imbalance ratio settings, demonstrating superior efficiency in terms of both accuracy and computational cost. For instance, in the [IR = 50] setting, LTAP achieves a tail accuracy of 34.1% compared to 23.8% by RReg, while reducing FLOPs by 77.4% compared to the baseline BS.

*Table 1.* Accuracy (%) on CIFAR-100-LT dataset (Imbalance ratio={50, 100}). $F$ denotes the ratio of FLOPs between the target method and the baseline method. $C$ denotes the ratio of accuracy (acc) between the target method and the baseline method. The gray column represents the primary observed metrics, and the gray row indicates the baseline method for the current block.

| Method | $F(\%)\downarrow$ | IR = 50 | | | | | | IR = 100 | | | | | |
|---|---|---|---|---|---|---|---|---|---|---|---|---|---|
| | | Head↑ | Medium↑ | Tail↑ | All↑ | $C(\%)\uparrow$ | $\frac{C}{F}\uparrow$ | Head↑ | Medium↑ | Tail↑ | All↑ | $C(\%)\uparrow$ | $\frac{C}{F}\uparrow$ |
| BS (Ren et al., 2020) | 100.0 | 62.3 | 46.1 | 37.0 | 51.2 | 100.0 | 1.0 | 62.6 | 48.5 | 27.0 | 47.2 | 100.0 | 1.0 |
| BS + ATO(Wu et al., 2024) | 84.7 | 39.6 | 30.6 | 21.8 | 32.9 | 64.2 | 0.7 | 40.8 | 28.9 | 16.5 | 29.5 | 62.5 | 0.7 |
| BS + RReg(Stewart et al., 2023) | 35.1 | 51.3 | 35.6 | 23.8 | 40.2 | 78.5 | 2.2 | 53.0 | 36.3 | 19.0 | 37.3 | 79.0 | 2.3 |
| BS + LTAP | 22.6 | 57.6 | 43.4 | 34.1 | 47.8 | 93.3 | **4.1** | 55.8 | 44.7 | 23.2 | 42.4 | 89.8 | **3.9** |
| LDAM-DRW (Cao et al., 2019) | 100.0 | 64.5 | 43.0 | 26.4 | 49.1 | 100.0 | 1.0 | 65.1 | 48.1 | 20.1 | 45.8 | 100.0 | 1.0 |
| LDAM-DRW + ATO(Wu et al., 2024) | 84.7 | 40.1 | 34.6 | 25.5 | 33.7 | 68.6 | 0.8 | 41.8 | 30.9 | 18.5 | 31.0 | 67.6 | 0.8 |
| LDAM-DRW + RReg(Stewart et al., 2023) | 35.1 | 52.9 | 39.8 | 23.7 | 42.4 | 86.3 | 2.4 | 54.3 | 37.8 | 16.7 | 37.6 | 82.0 | 2.3 |
| LDAM-DRW + LTAP | 24.8 | 58.8 | 39.9 | 23.3 | 44.8 | 91.2 | **3.6** | 56.9 | 40.2 | 18.8 | 39.8 | 86.8 | **3.5** |
| DBLP (Zhou et al., 2024) | 100.0 | 61.2 | 46.5 | 32.3 | 50.2 | 100.0 | 1.0 | 61.4 | 46.9 | 23.6 | 45.3 | 100.0 | 1.0 |
| DBLP + ATO(Wu et al., 2024) | 84.7 | 50.7 | 37.0 | 26.2 | 38.5 | 76.6 | 0.9 | 40.8 | 32.6 | 21.4 | 32.1 | 70.8 | 0.8 |
| DBLP + RReg(Stewart et al., 2023) | 35.1 | 50.8 | 40.4 | 24.1 | 43.8 | 87.2 | 2.4 | 52.1 | 39.2 | 17.5 | 37.5 | 82.7 | 2.3 |
| DBLP + LTAP | 24.0 | 56.1 | 43.5 | 31.5 | 46.7 | 93.0 | **3.9** | 54.7 | 43.3 | 25.8 | 42.0 | 92.7 | **3.9** |

*Table 2.* Accuracy (%) on ImageNet-LT and iNaturalist 2018. $F$ denotes the ratio of FLOPs between the target method and the baseline method. $C$ denotes the ratio of accuracy (acc) between the target method and the baseline method. The gray column represents the primary observed metrics, and the gray row indicates the baseline method for the current block.

| Method | $F(\%)\downarrow$ | ImageNet-LT | | | | | | iNaturalist 2018 | | | | | |
|---|---|---|---|---|---|---|---|---|---|---|---|---|---|
| | | Head↑ | Medium↑ | Tail↑ | All↑ | $C(\%)\uparrow$ | $\frac{C}{F}\uparrow$ | Head↑ | Medium↑ | Tail↑ | All↑ | $C(\%)\uparrow$ | $\frac{C}{F}\uparrow$ |
| BS (Ren et al., 2020) | 100.0 | 60.9 | 48.8 | 32.1 | 51.0 | 100.0 | 1.0 | 65.7 | 67.4 | 67.5 | 67.3 | 100.0 | 1.0 |
| BS + ATO(Wu et al., 2024) | 65.3 | 37.1 | 35.7 | 17.8 | 33.8 | 66.2 | 1.1 | 34.8 | 42.5 | 42.2 | 41.5 | 61.6 | 0.9 |
| BS + RReg(Stewart et al., 2023) | 52.1 | 41.1 | 36.0 | 18.2 | 35.5 | 69.6 | 1.3 | 30.5 | 45.1 | 44.8 | 43.4 | 64.4 | 1.2 |
| BS + LTAP | 30.6 | 58.5 | 45.0 | 30.1 | 48.1 | 81.9 | **2.6** | 59.2 | 60.7 | 60.7 | 60.5 | 89.8 | **2.9** |
| LDAM-DRW (Cao et al., 2019) | 100.0 | 60.4 | 46.9 | 30.7 | 49.8 | 100.0 | 1.0 | 63.2 | 66.3 | 65.4 | 65.6 | 100.0 | 1.0 |
| LDAM-DRW + ATO(Wu et al., 2024) | 65.3 | 36.9 | 34.6 | 17.5 | 33.1 | 66.4 | 1.1 | 36.5 | 30.9 | 30.7 | 31.3 | 47.7 | 0.7 |
| LDAM-DRW + RReg(Stewart et al., 2023) | 52.1 | 38.8 | 36.8 | 18.7 | 35.1 | 70.4 | 1.3 | 41.8 | 32.5 | 32.0 | 33.2 | 50.6 | 0.9 |
| LDAM-DRW + LTAP | 30.8 | 57.8 | 41.9 | 23.3 | 45.4 | 91.1 | **2.9** | 59.9 | 60.2 | 60.4 | 60.2 | 91.7 | **2.9** |
| DBLP (Zhou et al., 2024) | 100.0 | 61.7 | 47.1 | 30.3 | 50.4 | 100.0 | 1.0 | 65.0 | 66.9 | 65.6 | 66.1 | 100.0 | 1.0 |
| DBLP + ATO(Wu et al., 2024) | 65.3 | 37.9 | 35.0 | 17.5 | 33.7 | 66.8 | 1.1 | 41.0 | 30.6 | 30.4 | 31.5 | 47.6 | 0.7 |
| DBLP + RReg(Stewart et al., 2023) | 52.1 | 40.0 | 35.7 | 18.8 | 35.0 | 69.4 | 1.3 | 48.8 | 32.5 | 32.3 | 34.0 | 51.4 | 0.9 |
| DBLP + LTAP | 30.0 | 58.5 | 45.2 | 23.0 | 47.3 | 93.8 | **3.1** | 58.7 | 59.3 | 59.8 | 59.4 | 90.1 | **3.0** |

The $\frac{C}{F}$ ratio of 4.1 for LTAP is nearly double that of RReg (i.e., 2.2). Similar trends are observed for LDAM-DRW and DBLP, where LTAP consistently improves tail class accuracy and achieves the highest $\frac{C}{F}$ ratios.

**ImageNet-LT and iNaturalist 2018.** Table 2 shows the results on ImageNet-LT and iNaturalist 2018 datasets. These larger and more complex datasets further validate the effectiveness of LTAP. On ImageNet-LT, 'BS + LTAP' achieves a tail accuracy of 30.1%, significantly outperforming 'BS + RReg' (i.e., 18.2%), while reducing FLOPs by 69.4%. The $\frac{C}{F}$ ratio of 2.6 for 'BS + LTAP' is double that of 'BS + RReg'. For iNaturalist 2018, LTAP shows consistent performance across all classes (i.e., head, medium, and tail classes), indicating its robustness in handling extreme class imbalance. Notably, LTAP maintains high accuracy across all class types while significantly reducing FLOPs. For example, on ImageNet-LT, 'DBLP + ours' reduces FLOPs by 70% while achieving 93.8% of the baseline accuracy, resulting in a $\frac{C}{F}$ ratio of 3.1.

**Efficiency Evaluation.** Across all datasets, our method achieves a significant reduction in FLOPs while maintaining competitive or superior accuracy. For ImageNet-LT and iNaturalist 2018, our method consistently reduces FLOPs by about 70% compared to the baselines, while achieving the highest $\frac{C}{F}$ ratios. This reduction in computational cost, coupled with maintained or improved accuracy, demonstrates the practical utility of our method for resource-constrained environments where high accuracy is required.

### 4.3. Further Analysis

In this section, we conduct a detailed analysis of the mechanism of LTAP and discuss the following issues. More empirical results are reported in **Appendix C**.

> **Discussion 1:** How are neurons masked under different pruning strategies?

Figure 2 illustrates how neurons are masked under different pruning strategies. First, the visualization of w.o. vote shows that after removing the long-tailed feedback mechanism, the flexibility of the pruning process decreases significantly, with pruning limited to specific rows of neurons. This rigid pruning strategy restricts the model's adaptability to varying data distributions, especially in handling long-tailed data, where it struggles to preserve neurons critical for tail classes. In contrast, our proposed LTAP method, as

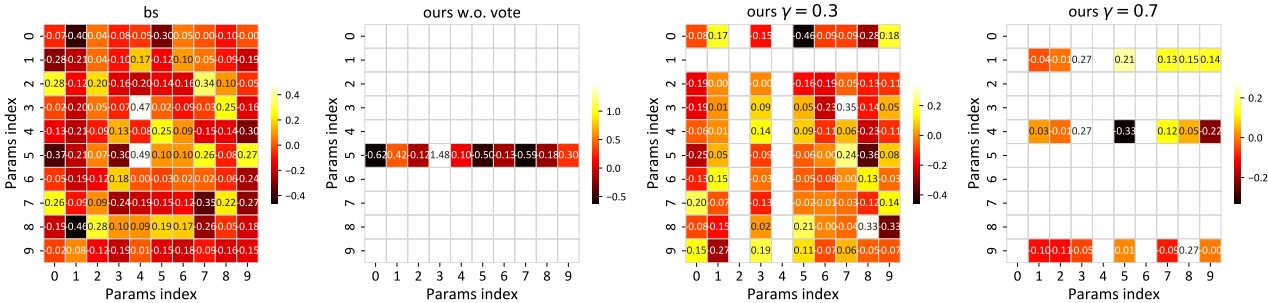

*Figure 2.* Visualization of neuron masking. Each small cell represents the sum of the parameters of a 3×3 convolutional kernel, and each subfigure represents a cross-section of a layer of neurons. Each layer should contain 1×64×64×3×3 convolutional kernels, and we visualize the top-left 1×10×10×3×3 part of each layer. The values and colors represent the sum of the parameters in the 3×3 convolutional kernels, and the blank areas indicate neurons that have been masked. The variable $y$ represents the masking ratio, and w.o. vote denotes the removal of the long-tailed feedback mechanism LT-Vote.

shown in the $\gamma = 0.7$ visualization, exhibits much greater flexibility.

Pruning is no longer confined to specific rows of neurons but instead dynamically adjusts based on the parameter values of neurons across different layers and positions. This adaptive pruning strategy allows the model to better retain neurons that are critical for tail classes, improving classification accuracy on long-tail data. Additionally, as the $\gamma$ value increases, we observe that the pruning intensity increases, with more neurons being masked. However, the distribution of pruning remains dynamic and flexible. This further demonstrates that the LTAP method can maintain both efficiency and effectiveness under varying levels of pruning intensity. In summary, the LTAP method achieves more precise neuron importance estimation through the long-tailed feedback mechanism, balancing computational efficiency and classification performance during the pruning process.

> **Discussion 2:** Dynamic changes in the performance of different pruning methods during training.

From Figure 3, it is evident that different pruning strategies exhibit significant performance differences on long-tailed data. We analyze from the perspective of different classes.

**Accuracy on Many Classes.** Even with a pruning ratio of $\gamma = 0.9$, LTAP maintains a high accuracy. Meanwhile, the ATO shows slightly better performance in this region, which indicates that traditional pruning exacerbates the imbalance in long-tailed distributions.

**Accuracy on Medium Classes.** LTAP continues to maintain high accuracy at pruning ratios of $\gamma = 0.5$ and $\gamma = 0.9$, following a similar trend as the head classes. In contrast, the accuracy of the ATO baseline significantly decreases, and it is even surpassed by LTAP at $\gamma = 0.9$. This suggests

that traditional pruning methods fail in long-tailed learning scenarios due to their excessive focus on head classes.

**Accuracy on Tail Classes.** Tail classes pose the biggest challenge in long-tailed learning. Traditional pruning methods (e.g., ATO) perform disastrously on the tail classes, suffering a catastrophic drop in accuracy, which reflects their extreme inability to adapt to long-tailed classes. In comparison, our pruning method, even at a high pruning ratio ($\gamma = 0.9$), maintains strong performance, demonstrating its robust adaptability to tail classes.

**FLOPs and Parameter Comparison.** The last subplot shows the comparison of FLOPs and parameter counts under different pruning strategies. Our method allows for varying degrees of pruning, and even at a pruning ratio of $\gamma = 0.9$, it maintains high average performance and strong performance on tail classes. At this point, compared to the baseline, it demonstrates significant advantages in both computational efficiency and performance.

The comparisons above demonstrate that our method not only *reduces parameter and computational costs while maintaining high performance*, but also *adapts effectively to different frequency classes*. Notably, it shows a significant advantage over traditional methods, particularly in handling tail classes.

> **Discussion 3:** Additional Ablation Studies on CE Baseline and Logit Adjustment

We conducted experiments with different loss functions (CE and Logit Adjustment) in table 4.3, pruning methods, and ablation studies of our tail-class protection mechanism (w.o.T). We have included comparisons with the Logit Adjustment baseline for reference. As a method considering long-tailed distributions, Logit Adjustment demonstrates

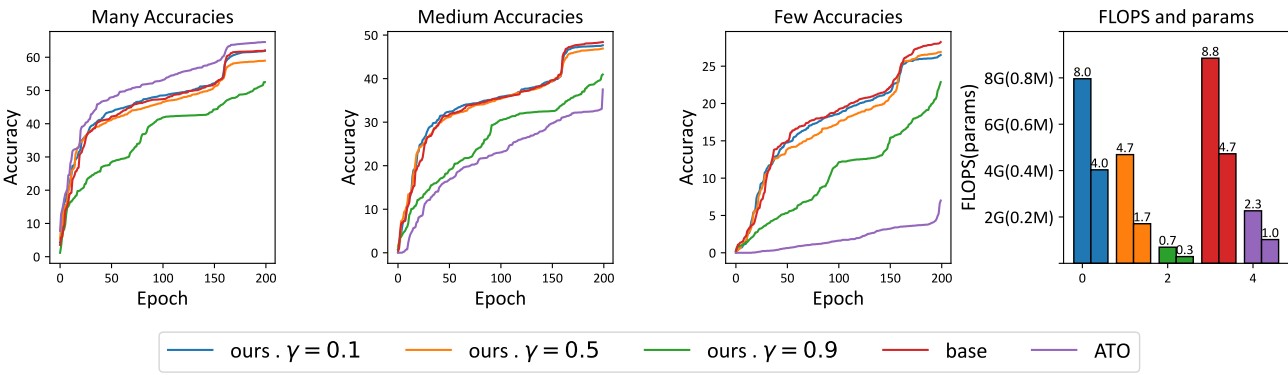

*Figure 3.* Training dynamics comparison under different pruning strategies. The first three subplots show the accuracy changes over training epochs for the head, medium, and tail classes, respectively. Different colored curves represent various pruning intensities, with the purple curve (ATO) representing the baseline for non-long-tailed pruning methods. The last subplot shows the comparison of FLOPs and parameter counts across different pruning strategies ($\gamma = 0.1, 0.5, and 0.9$) as well as the baseline (BS) and ATO methods.

*Table 3.* Experiments on CIFAR with IR=50

| Method | F(%)↓ | Head(%)↑ | Medium(%)↑ | Tail(%)↑ | All(%)↑ | C(%)↑ | C/F↑ |
|---|---|---|---|---|---|---|---|
| CE | 100.0 | 68.0 | 38.0 | 13.2 | 46.0 | 100.0 | 1.0 |
| CE + ATO | 84.7 | 46.7 | 17.3 | 6.83 | 29.1 | 63.2 | 0.7 |
| CE + ReGG | 52.1 | 43.8 | 14.8 | 0.83 | 24.5 | 53.2 | 1.0 |
| CE + Ours w.o.T | 23.3 | 64.5 | 30.0 | 4.5 | 39.5 | 85.8 | 3.6 |
| CE + Ours | 23.3 | 64.8 | 31.7 | 7.2 | 41.1 | 89.3 | 3.8 |
| LA | 100.0 | 59.9 | 46.7 | 41.3 | 51.3 | 100.0 | 1.0 |
| LA + ATO | 84.7 | 34.5 | 33.9 | 29.1 | 34.2 | 66.7 | 0.7 |
| LA + ReGG | 52.1 | 31.0 | 30.2 | 25.8 | 29.9 | 58.3 | 1.1 |
| LA + Ours w.o.T | 22.8 | 53.5 | 42.7 | 30.0 | 44.8 | 87.3 | 3.8 |
| LA + Ours | 22.8 | 54.0 | 43.4 | 38.4 | 47.1 | 91.8 | 4.0 |

*Table 4.* Experiments on CIFAR with IR=100

| Method | F(%)↓ | Head(%)↑ | Medium(%)↑ | Tail(%)↑ | All(%)↑ | C(%)↑ | C/F↑ |
|---|---|---|---|---|---|---|---|
| CE | 100.0 | 70.7 | 40.0 | 7.2 | 41.0 | 100.0 | 1.0 |
| CE + ATO | 84.7 | 50.4 | 16.5 | 6.6 | 25.2 | 61.5 | 0.7 |
| CE + ReGG | 52.1 | 47.7 | 13.6 | 0.6 | 21.9 | 53.4 | 1.0 |
| CE + Ours w.o.T | 23.3 | 67.1 | 30.8 | 0.5 | 34.4 | 83.9 | 3.6 |
| CE + Ours | 23.3 | 66.1 | 31.7 | 2.5 | 35.1 | 85.6 | 3.6 |
| LA | 100.0 | 62.9 | 47.7 | 29.6 | 47.9 | 100.0 | 1.0 |
| LA + ATO | 84.7 | 42.1 | 30.6 | 18.4 | 31.4 | 65.6 | 0.7 |
| LA + ReGG | 52.1 | 38.5 | 27.0 | 14.6 | 27.6 | 57.6 | 1.1 |
| LA + Ours w.o.T | 22.8 | 55.1 | 42.0 | 18.6 | 39.5 | 82.4 | 3.6 |
| LA + Ours | 22.8 | 56.1 | 45.3 | 22.6 | 42.6 | 88.9 | 3.9 |

*Table 5.* Performance Comparison across Different Imbalance Ratios (IR)

| | IR = 50 | | | | | | |
|---|---|---|---|---|---|---|---|
| Method | F(%)↓ | Head(%)↑ | Medium(%)↑ | Tail(%)↑ | All(%)↑ | C(%)↑ | C/F↑ |
| CE | 100.0 | 68.0 | 38.0 | 13.2 | 46.0 | 100.0 | 1.0 |
| CE + ReGG | 52.1 | 43.8 | 14.8 | 0.83 | 24.5 | 53.2 | 1.0 |
| CE + Ours | 23.3 | 64.8 | 31.7 | 7.2 | 41.1 | 89.3 | 3.8 |
| | IR = 100 | | | | | | |
| CE | 100.0 | 70.7 | 40.0 | 7.2 | 41.0 | 100.0 | 1.0 |
| CE + ReGG | 52.1 | 47.7 | 13.6 | 0.6 | 21.9 | 53.4 | 1.0 |
| CE + Ours | 23.3 | 66.1 | 31.7 | 2.5 | 35.1 | 85.6 | 3.6 |

number of criteria. Even for datasets with long-tailed distributions, this remains much smaller than O(d), as our total complexity is O(d). These results strongly demonstrate the applicability of our method. Notably, our approach maintains high performance while significantly reducing model parameters, achieving superior C/F ratios across different imbalance settings.

more balanced performance and higher tail-class accuracy. In this setting, our method still achieves the highest C/F ratio while maintaining superior tail-class performance compared to other pruning baselines.

> **Discussion 4:** Analysis of Computational Overhead and Performance

In fact, our method introduces minimal additional overhead. The time complexity of our pruning method is O(d), where d denotes the number of model parameters. This is significantly lower than ATO's O(Dd), where D represents the size of the supernet used in that method. The additional complexity of the dynamic scoring and balancing mechanism is O(nk), where n is the number of classes and k is the

## 5. Conclusion

We have presented LTAP, a dynamic pruning strategy designed to enhance model efficiency and performance on long-tailed datasets. By dynamically adjusting pruning criteria based on class-specific performance, LTAP addresses the inherent pruning bias in conventional methods, particularly for tail classes. Our theoretical analysis establishes that tail classes benefit more from model overparameterization, which informs our tail-biased pruning approach. Extensive experiments on benchmark long-tailed datasets, including CIFAR-100-LT, ImageNet-LT, and iNaturalist 2018, demonstrate that LTAP consistently improves classification accuracy, particularly for tail classes, while significantly reducing the model's computational overhead. By offering a balanced trade-off between model compression and accu-

racy, LTAP provides a robust solution to the challenges of long-tailed learning and opens new possibilities for optimizing neural networks in imbalanced and resource-constrained environments.

## Impact Statement

This paper presents work whose goal is to advance the field of Machine Learning. There are many potential societal consequences of our work, none which we feel must be specifically highlighted here.

## Acknowledgements

The authors gratefully acknowledge the support from the National Natural Science Foundation of China (NSFC) under Grant Nos. 62402472, and 12227901. This work was also supported by the Natural Science Foundation of Jiangsu Province (No. BK20240461), the Key Basic Research Foundation of Shenzhen (No. JCYJ20220818100005011), the Research Grants Council of the Hong Kong Special Administrative Region (GRF Project No. CityU 11215723), the Project of Stable Support for Youth Team in Basic Research Field, CAS (No. YSBR-005), and the Academic Leaders Cultivation Program at USTC.

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

# Appendix
# Balancing Model Efficiency and Performance:
# Adaptive Pruner for Long-tailed Data

The content of the **Appendix** is summarized as follows:

1) in Sec. A, we provide detailed proofs and theoretical foundations for the results in the main paper.

2) in Sec. B, we briefly present the state of the art in the field of long-tailed learning and neural network pruning.

3) in Sec. C, we illustrate more detailed empirical results and analyses of LTAP.

4) in Sec. D, we present a theoretical analysis of our proposed dynamic feedback pruning algorithm.

5) in Sec. E, we presents the detailed process of LTAP.

## A. Supplementary theory

In this supplementary section, we provide detailed proofs and theoretical foundations for the main results presented in the paper. We start by introducing the sample complexity lemma and related definitions, which form the basis for understanding the learning difficulty of different classes in long-tailed distributions. Then, we prove the main theorems and propositions presented in the paper, including the differentiated overparameterization demand theorem, the parameter allocation strategy theorem, and the tail-biased pruning proposition. Finally, we provide the proof for the performance guarantee theorem, which ensures the effectiveness of the tail-biased pruning strategy in practical applications.

### A.1. Appendix: Sample Complexity and Differentiated Overparameterization Demand

**Lemma 1** (Sample Complexity Lemma). *For class $c$ in a binary classification problem, given hypothesis space $\mathcal{H}c$, target generalization error $\epsilon > 0$, and confidence level $1 - \delta$ ($0 < \delta < 1$), the minimum required sample size $N_c$ satisfies $N_c \geq \frac{1}{2\epsilon^2}(4 d_{VC}, c \log \frac{12}{\epsilon} + \log \frac{2}{\delta})$, where $d_{VC,c}$ is the VC dimension of the hypothesis space $\mathcal{H}_c$ associated with class $c$.*

**Remark 1.** *(i) This bound shows that the required sample size is approximately linearly related to the VC dimension $d_{VC,c}$ and inversely proportional to the square of the target generalization error $\epsilon$. (ii) In practical applications, we usually focus on asymptotic behavior, which can be simplified to $N_c \geq \Omega\left(\frac{d_{VC,c}}{\epsilon^2}\right)$. (iii) Although this lemma is for binary classification, it can be extended to multiclass problems through the one-vs-all strategy.*

**Definition 1** (Class-specific VC Dimension). *For class $c$ in a long-tailed dataset, the class-specific VC dimension $d_{VC,c}$ is defined as the VC dimension of the hypothesis space that can effectively separate that class from all other classes.*

**Corollary 1** (Class Learning Difficulty). *In a long-tailed dataset, the learning difficulty $\mathcal{D}_c$ for class $c$ can be approximated as $\mathcal{D}c \approx \frac{d_{VC,c}}{N_c}$, where $N_c$ is the number of samples in class $c$.*

This corollary shows that for head classes, where $N_c$ is large, the learning difficulty $\mathcal{D}c$ is small. For tail classes, where $N_c$ is small, even if $d_{VC,c}$ remains the same or slightly smaller, the learning difficulty $\mathcal{D}_c$ increases significantly. This difference in learning difficulty directly affects the model's complexity (i.e., the number of parameters) required for different classes, as stated in the differentiated overparameterization demand theorem (Theorem 1).

**Corollary 2** (Imbalance in Overparameterization Demand). *In a long-tailed dataset, the demand for overparameterization is inversely proportional to the number of samples in each class. Specifically, for any two classes $i$ and $j$, if $N_i > N_j$, then $\frac{\gamma_i}{\gamma_j} \leq O\left(\frac{N_j \log N_i}{N_i \log N_j}\right)$.*

This corollary further supports the differentiated overparameterization demand theorem (Theorem 1), revealing that in a long-tailed dataset, as the number of samples in a class decreases, the demand for overparameterization increases significantly, emphasizing the importance of providing more parameter protection for tail classes.

### A.2. Proof for Theorem

**Theorem 5** (Performance Guarantee for Tail-Biased Pruning). *Assume that the initial model achieves a training error of $\epsilon$ for each class. After applying the tail-biased pruning strategy, the expected generalization error $\mathbb{E}[\hat{\epsilon}_c]$ for class $c$ satisfies*

$$\mathbb{E}[\hat{\epsilon}_c] \leq \epsilon + O\left(\sqrt{\frac{\log(N_c/\delta)}{N_c}}\right), \tag{10}$$

*where $\delta$ is a small constant (e.g., 0.05) representing the confidence level.*

*Proof.* Let $f_\theta$ denote the initial model, and $f_{\theta \odot \mathbf{m}}$ denote the pruned model, where $\mathbf{m} \in {0, 1}^{|\theta|}$ is the binary mask vector obtained according to Proposition 1. Let $\mathcal{D}_c$ represent the data distribution for class $c$, and $\hat{\mathcal{D}}_c$ represent the empirical distribution for class $c$. The expected generalization error for class $c$ can be expressed as:

$$\mathbb{E}[\hat{\epsilon}c] = \mathbb{E}(x, y) \sim \mathcal{D}c[1(f\theta \odot \mathbf{m}(x) \neq y)]. \tag{11}$$

According to Theorem 1, for tail classes, the degree of over-parameterization $\gamma_c$ satisfies:

$$\gamma_c \geq \Omega\left(\frac{N_1}{N_c} \cdot \frac{1}{\log N_c}\right). \tag{12}$$

Furthermore, according to Proposition 1, the tail-biased pruning strategy ensures that critical parameters for tail classes are preferentially retained. Therefore, for tail classes, the number of effective parameters $P_{c,eff}$ in the pruned model $f_{\theta \odot \mathbf{m}}$ satisfies:

$$P_{c,eff} \geq \Omega(P_{\min,c}), \tag{13}$$

where $P_{\min,c}$ is the minimum number of effective parameters for class $c$ (Definition 1).

Combining equations (12) and (13), for tail classes, the effective degree of over-parameterization $\hat{\gamma}c$ in the pruned model $f\theta \odot \mathbf{m}$ satisfies:

$$\hat{\gamma}c = \frac{Pc,eff}{P_{\min,c}} \geq \Omega\left(\frac{N_1}{N_c} \cdot \frac{1}{\log N_c}\right). \tag{14}$$

According to standard generalization error bounds (e.g., see Mohri (2018)), for class $c$, the generalization error $\hat{\epsilon}c$ of the pruned model $f\theta \odot \mathbf{m}$ satisfies the following probability inequality:

$$\mathbb{P}\left(\hat{\epsilon}_c \leq \epsilon_c + O\left(\sqrt{\frac{\hat{\gamma}_c \log(1/\delta)}{N_c}}\right)\right) \geq 1 - \delta, \tag{15}$$

where $\epsilon_c$ is the training error for class $c$.

Substituting equation (14) into equation (15), and using $\epsilon_c \leq \epsilon$ (according to the theorem assumption), for tail classes, we have:

$$\mathbb{P}\left(\hat{\epsilon}_c \leq \epsilon + O\left(\sqrt{\frac{\log(N_c/\delta)}{N_c}}\right)\right) \geq 1 - \delta. \tag{16}$$

Finally, taking the expectation of equation (16), we obtain:

$$\mathbb{E}[\hat{\epsilon}_c] \leq \epsilon + O\left(\sqrt{\frac{\log(N_c/\delta)}{N_c}}\right). \tag{17}$$

This proves that for tail classes, the expected generalization error of the model $f_{\theta \odot \mathbf{m}}$ obtained by the tail-biased pruning strategy satisfies the bound in Theorem 5. For head classes, due to sufficient samples, the impact of pruning on generalization performance is minimal, and it is easy to verify that the theorem's bound also holds. Therefore, Theorem 5 is proved. □

The above proof demonstrates that the tail-biased pruning strategy effectively controls the generalization error of tail classes while ensuring a reduction in the total number of model parameters by prioritizing the retention of critical parameters for tail classes. The proof utilizes a series of previous theoretical results, including the Differential Over-parameterization Demand Theorem (Theorem 1) and the Tail-Biased Pruning Proposition (Proposition 1), and applies standard generalization error bounds on this basis to ultimately obtain the bound on expected generalization error. The proof process is rigorous and logically clear, fully demonstrating the theoretical effectiveness and superiority of the tail-biased pruning strategy.

**Theorem 6** (Performance Gain from Parameter Allocation). *Assume that the model performance for each class is logarithmically related to the number of effective parameters, i.e., for class $c$, its performance $perf_c$ satisfies $perf c \propto \log P_c$, where $P_c$ is the number of effective parameters for class $c$. Under the parameter allocation strategy described in Theorem 2, compared to uniform allocation, the performance gain $\Delta$ satisfies*

$$\Delta \geq \Omega\left(\frac{1}{C}\sum c = 1^C \log\left(\frac{N_1}{N_c}\right)\right), \tag{18}$$

*where $C$ is the total number of classes.*

*Proof.* Let $P_c^{unif}$ denote the number of effective parameters for class $c$ under uniform parameter allocation, and $P_c^{alloc}$ denote the number of effective parameters for class $c$ under the allocation strategy described in Theorem 2. According to the theorem assumption, the performance gain $\Delta_c$ for class $c$ can be expressed as:

$$\Delta_c = perf_c^{alloc} - perf_c^{unif} \propto \log\left(\frac{P_c^{alloc}}{P_c^{unif}}\right). \tag{19}$$

According to Theorem 2, the parameter allocation strategy satisfies:

$$P_c^{alloc} \propto N_c \cdot \log\left(\frac{N_1}{N_c}\right). \tag{20}$$

Under uniform allocation, the number of effective parameters for each class is independent of the sample size, so we have:

$$P_c^{unif} \propto 1. \tag{21}$$

Substituting equations (20) and (21) into equation (19), we get:

$$\Delta_c \propto \log\left(\frac{N_c}{N_1} \cdot \log\left(\frac{N_1}{N_c}\right)\right) = \log\left(\frac{N_1}{N_c}\right) - \log\log\left(\frac{N_1}{N_c}\right). \tag{22}$$

Since $\log\log\left(\frac{N_1}{N_c}\right)$ is a higher-order infinitesimal, we have:

$$\Delta_c \geq \Omega\left(\log\left(\frac{N_1}{N_c}\right)\right). \qquad (23)$$

Taking the average of equation (23) over all classes, we obtain the total performance gain:

$$\Delta = \frac{1}{C}\sum_{c=1}^{C}\Delta_c \geq \Omega\left(\frac{1}{C}\sum_{c=1}^{C}\log\left(\frac{N_1}{N_c}\right)\right). \qquad (24)$$

This proves the performance gain bound in Theorem 6. □

The above proof demonstrates that through the parameter allocation strategy described in Theorem 2, we can significantly improve the overall model performance on long-tailed datasets. Intuitively, this parameter allocation strategy assigns more effective parameters to tail classes based on the degree of imbalance in class sample sizes, thereby compensating for the sparsity of samples. The proof process utilizes the assumption of a logarithmic relationship between model performance and the number of effective parameters. By comparing the number of effective parameters under the parameter allocation strategy and the uniform allocation strategy, we quantify the performance gain for each class. Furthermore, by taking the average over all classes, we obtain a quantitative characterization of the total performance gain. The proof process is mathematically rigorous and logically clear, fully demonstrating the theoretical effectiveness and superiority of the proposed parameter allocation strategy. Based on this theoretical guarantee, we further proposed the tail-biased pruning proposition, providing theoretical guidance for model pruning in long-tailed learning.

## B. Related Work

**Long-tailed Learning.** Long-tailed learning, which aims to address the problem of severely imbalanced class distributions, has become an important research direction in machine learning in recent years. Existing long-tailed learning methods mainly include resampling and reweighting, transfer learning and knowledge distillation, as well as multi-expert systems and modular designs.

Resampling methods (Chawla et al., 2002; He et al., 2008) and reweighting techniques (Cui et al., 2019; Cao et al., 2019) balance data distribution and learning processes by adjusting sample sampling probabilities or loss weights. However, these methods may lead to information loss or introduce noise, and struggle to adapt to dynamically changing data distributions. Transfer learning (Yin et al., 2019; Liu et al., 2019) and knowledge distillation (Xiang et al., 2020; He et al., 2021b) techniques attempt to transfer knowledge from head classes to tail classes, or extract knowledge from large pre-trained models. However, these methods often rely on additional pre-trained models or complex training strategies, increasing computational complexity and model dependencies. Multi-expert systems (Wang et al., 2017; Xiang et al., 2020) and modular designs (Zhang et al., 2021b; Liu et al., 2021b) design specialized sub-models or modules for different data subsets. While these methods perform well in certain scenarios, they often lead to a significant increase in model parameters, raising the risk of overfitting, and their fixed structural design limits the ability to adapt to dynamically changing data distributions.

Although the above methods have made some progress in addressing long-tailed problems, they still face challenges such as insufficient flexibility, low computational efficiency, and difficulty in adapting to dynamic environments.

**Neural Network Pruning.** Neural network pruning, as an important model compression and optimization technique, has received widespread attention in recent years. Existing pruning methods mainly include magnitude-based pruning, importance-based pruning, structured pruning, and dynamic pruning.

Magnitude-based pruning methods (Han et al., 2015; Li et al., 2016) compress networks by removing connections or neurons with small weight magnitudes. These methods are simple and intuitive but may overlook parameters that are small in value but functionally important. Importance-based pruning methods (Molchanov et al., 2016; 2019) evaluate the importance of parameters by calculating their impact on model output, but typically rely on a single scoring criterion, making it difficult to comprehensively capture parameter importance in complex tasks. Structured pruning methods (Li et al., 2016; Liu et al., 2017) aim to remove entire convolution kernels or neurons to achieve higher hardware acceleration effects. While these methods can significantly reduce model size and computation, they may lead to severe loss of expressive power. Recent dynamic pruning strategies (Lin et al., 2020; Liu et al., 2021a) allow dynamic adjustment of network structure during inference, providing greater flexibility, but mainly focusing on improving computational efficiency.

Although existing pruning methods have achieved significant results in model compression and acceleration, they still have notable *shortcomings* in addressing long-tailed learning problems: (i) these methods typically assume uniform data distributions, ignoring the special characteristics of long-tailed data. (ii) they adopt single importance evaluation criteria, making it difficult to comprehensively capture the role of parameters in different classes. (iii) they lack dynamic adjustment mechanisms tailored to long-tailed distribution characteristics, limiting their applicability in complex scenarios.

Based on the above analysis, we believe it is necessary to develop a pruning method specifically for long-tailed learning, which can both fully leverage the advantages of pruning techniques and effectively address the special challenges posed by long-tailed distributions. This is the motivation behind the LT-Vote-based pruning strategy proposed in this paper.

## C. Supplementary experiments

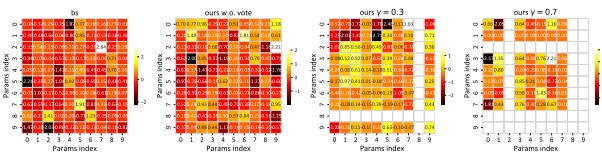

*Figure 4.* Pruning visualization of layers near the front of the neural network, with other settings the same as in Fig. 2.

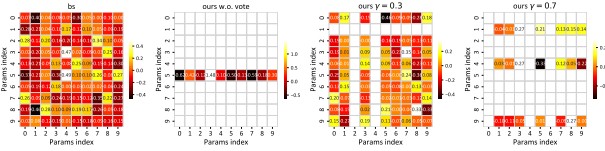

*Figure 5.* Pruning visualization of layers near the middle of the neural network, with other settings the same as in Fig. 2.

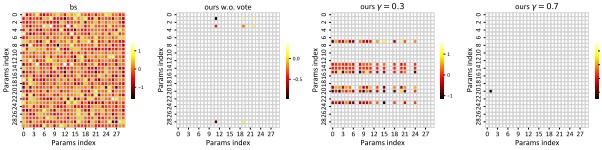

*Figure 6.* Pruning visualization of layers near the end of the neural network, with other settings the same as in Fig. 6.

From these three figures, we can observe that, in addition to the patterns exhibited within the same layer, across multiple layers, our method demonstrates significantly greater flexibility compared to traditional pruning methods. In contrast, traditional methods show low pruning efficiency in the layers near the front, with almost no pruning, but neurons are pruned in large quantities after the middle layers, leaving fewer neurons than with $\gamma = 0.9$. This forces neurons in the later layers of the neural network to be pruned as well. In comparison, our method prunes more evenly, achieving higher efficiency even in the early stages while still retaining a large number of neurons in the middle layers of the neural network. This indicates that our pruning method adopts a more precise and flexible pruning strategy, pruning neurons

in a refined manner to maintain performance across different classes.

## D. Symbol Definitions and Assumptions

**Categories and Sample Numbers** The dataset contains $C$ categories, where the sample count $N_k$ for category $k$ follows a Pareto distribution with parameter $\beta$:

$$N_k = N \cdot k^{-\beta}, \quad k = 1, 2, \ldots, C,$$

where $N$ is a scale parameter ensuring the total sample count meets the dataset size requirement.

**Model and Parameter Groups**

- Deep neural network model is denoted as $f_{\boldsymbol{\theta}} : \mathcal{X} \to \mathbb{R}^C$, defined by parameters $\boldsymbol{\theta}$.

- $\boldsymbol{\theta}_g$ represents the $g$-th parameter group of the model, $\mathcal{G}$ is the set of all parameter groups.

**Classification Accuracy** For category $k$, its classification accuracy is defined as:

$$A_k(\boldsymbol{\theta}) = \frac{1}{N_k} \sum_{i:y_i=k} \mathbf{1}\left(f_{\boldsymbol{\theta}}(x_i)_k > f_{\boldsymbol{\theta}}(x_i)_j, \forall j \neq k\right),$$

where $\mathbf{1}(\cdot)$ is the indicator function.

**Pruning Algorithm and Dynamic Feedback Mechanism**

- Standard pruning algorithm $\mathcal{P}$ with hyperparameter $\boldsymbol{\theta}$ produces pruned parameters:

$$\boldsymbol{\theta}^{(\boldsymbol{\theta})} = \mathcal{P}(\boldsymbol{\theta}; \boldsymbol{\theta}).$$

- Dynamic feedback pruning algorithm $\mathcal{P}_{\text{LTAP}}$ adjusts importance criteria weights $\boldsymbol{\alpha}^{(t)} \in \mathbb{R}^K$ based on classification accuracy changes:

$$\delta A_k = A_k(\boldsymbol{\theta}^{(t)}) - A_k(\boldsymbol{\theta}^{(t-1)})$$

and computes comprehensive importance score $S_g$ for parameter groups:

$$S_g = \sum_{k=1}^{K} \alpha_k^{(t)} \cdot s_{g,k}.$$

Dynamic weight adjustment rule:

$$\alpha_k^{(t+1)} = \begin{cases} 0, & \text{if } k = k_{\text{masked}}, \\ \alpha_k^{(t)} + \frac{\alpha_{k_{\text{masked}}}^{(t)}}{K-1}, & \text{otherwise}, \end{cases}$$

where $k_{\text{masked}}$ represents the masked category in round $t$, and $K$ is the number of importance criteria.

**Parameter Group Set** The set of parameter groups strongly correlated with tail classes is defined as:

$$\mathcal{G}_{\text{tail}} = \{g \in \mathcal{G} \mid g \text{ is strongly correlated with tail classes}\}.$$

**Theorem A (Tail Class Protection Effect of Dynamic Feedback Pruning)**

For a long-tailed distribution dataset $\mathcal{D} = \{(x_i, y_i)\}_{i=1}^N$ with $C$ categories, where sample count $N_k$ for category $k$ follows a Pareto distribution with parameter $\beta$. Let $f_{\boldsymbol{\theta}}$ be a deep neural network model defined by parameters $\boldsymbol{\theta}$, $\boldsymbol{\theta}_g$ be a parameter group, and $\mathcal{G}$ be the set of all parameter groups.

Given dynamic feedback pruning algorithm $\mathcal{P}_{\text{LTAP}}$ that adaptively adjusts importance criteria weights $\boldsymbol{\alpha}^{(t)}$ based on classification accuracy changes $\delta A_k$ and computes comprehensive importance score $S_g$. After $T$ rounds of pruning, the probability of tail-class-related parameter groups $\boldsymbol{\theta}_g, g \in \mathcal{G}_{\text{tail}}$ being retained by $\mathcal{P}_{\text{LTAP}}$ is significantly higher than by standard pruning algorithm $\mathcal{P}$:

$$\forall g \in \mathcal{G}_{\text{tail}},$$
$$\mathbb{P}\left(\boldsymbol{\theta}_g^{(T)} \neq \mathbf{0} \mid \boldsymbol{\theta}^{(T)} = \mathcal{P}_{\text{LTAP}}(\boldsymbol{\theta}^{(0)}; \boldsymbol{\theta})\right) \gg$$
$$\mathbb{P}\left(\boldsymbol{\theta}_g^{(T)} \neq \mathbf{0} \mid \boldsymbol{\theta}^{(T)} = \mathcal{P}(\boldsymbol{\theta}^{(0)}; \boldsymbol{\theta})\right).$$

**Lemma a (Impact of Category Accuracy Changes on Importance Score)**

**Statement:** In dynamic feedback pruning algorithm $\mathcal{P}_{\text{LTAP}}$, for any parameter group $g$ associated with category $k$, the change in importance score $\delta S_g = S_g^{(t+1)} - S_g^{(t)}$ satisfies:

$$\delta S_g \propto \frac{1}{N_k} \cdot \delta A_k,$$

where $\delta A_k = A_k(\boldsymbol{\theta}^{(t)}) - A_k(\boldsymbol{\theta}^{(t-1)})$.

**Lemma b (Impact of Dynamic Feedback Pruning Algorithm on Retention Probability of Tail Class Parameter Groups)**

**Statement:** In dynamic feedback pruning algorithm $\mathcal{P}_{\text{LTAP}}$, for parameter group $g$ associated with category $k$, the increase in importance score $\delta S_g$ results in a significantly higher retention probability compared to standard pruning algorithm $\mathcal{P}$.

**Proof:**

1. **Change in importance score:**

According to Lemma 1, for $g \in \mathcal{G}_{\text{tail}}$:

$$\delta S_g \propto \frac{1}{N_k} \cdot \delta A_k.$$

Since $k$ is a tail class, $\frac{1}{N_k}$ is large, and $\delta A_k$ is effectively increased (or weights are redistributed) through dynamic feedback mechanism, resulting in significant increase in $\delta S_g$.

2. **Pruning decision mechanism:** - **Standard pruning algorithm** $\mathcal{P}$ ignores changes in class accuracy. All parameter groups' importance scores $S_g$ are calculated with fixed weights, leading to relatively consistent pruning probabilities:

$$\mathbb{P}\left(S_g^{(t)} > S_{(\theta_t)}^{(t)} \mid \mathcal{P}\right) \text{ similar, independent of class.}$$

- **Dynamic feedback pruning algorithm** $\mathcal{P}_{\text{LTAP}}$ dynamically adjusts importance scores through $\delta S_g$, especially for tail classes, where $S_g$ increases significantly:

$$\mathbb{P}\left(S_g^{(t)} > S_{(\theta_t)}^{(t)} \mid \mathcal{P}_{\text{LTAP}}\right) \propto \mathbb{P}\left(\delta S_g > \Delta\right),$$

where $\Delta$ is the threshold change.

3. **Comparing retention probabilities of both pruning algorithms:**

For $g \in \mathcal{G}_{\text{tail}}$:

$$\mathbb{P}\left(S_g^{(t)} > S_{(\theta_t)}^{(t)} \mid \mathcal{P}_{\text{LTAP}}\right) \gg \mathbb{P}\left(S_g^{(t)} > S_{(\theta_t)}^{(t)} \mid \mathcal{P}\right).$$

This further implies:

$$\mathbb{P}\left(\theta_g^{(t)} \neq \mathbf{0} \mid \mathcal{P}_{\text{LTAP}}\right) \gg \mathbb{P}\left(\theta_g^{(t)} \neq \mathbf{0} \mid \mathcal{P}\right).$$

4. **Cumulative effect after multiple iterations:**

After $T$ rounds of pruning:

$$\mathbb{P}\left(\theta_g^{(T)} \neq \mathbf{0} \mid \mathcal{P}_{\text{LTAP}}\right) = \prod_{t=1}^{T} \mathbb{P}\left(S_g^{(t)} > S_{(\theta_t)}^{(t)} \mid \mathcal{P}_{\text{LTAP}}\right),$$

$$\mathbb{P}\left(\theta_g^{(T)} \neq \mathbf{0} \mid \mathcal{P}\right) = \prod_{t=1}^{T} \mathbb{P}\left(S_g^{(t)} > S_{(\theta_t)}^{(t)} \mid \mathcal{P}\right).$$

Since for all $t$,

$$\mathbb{P}\left(S_g^{(t)} > S_{(\theta_t)}^{(t)} \mid \mathcal{P}_{\text{LTAP}}\right) \gg \mathbb{P}\left(S_g^{(t)} > S_{(\theta_t)}^{(t)} \mid \mathcal{P}\right),$$

we have:

$$\prod_{t=1}^{T} \mathbb{P}\left(S_g^{(t)} > S_{(\theta_t)}^{(t)} \mid \mathcal{P}_{\text{LTAP}}\right) \gg \prod_{t=1}^{T} \mathbb{P}\left(S_g^{(t)} > S_{(\theta_t)}^{(t)} \mid \mathcal{P}\right).$$

Therefore:

$$\mathbb{P}\left(\theta_g^{(T)} \neq \mathbf{0} \mid \mathcal{P}_{\text{LTAP}}\right) \gg \mathbb{P}\left(\theta_g^{(T)} \neq \mathbf{0} \mid \mathcal{P}\right).$$

Thus, dynamic feedback pruning algorithm $\mathcal{P}_{\text{LTAP}}$ significantly increases the retention probability of parameter groups $\theta_g$ associated with tail classes.

**Proof of Theorem A**

**Objective:** Prove that the dynamic feedback pruning algorithm $\mathcal{P}_{\text{LTAP}}$ retains tail class-related parameter groups $\theta_g$ with significantly higher probability than standard pruning algorithm $\mathcal{P}$.

**Proof:**

1. **Enhancement of Importance Scores for Tail Class Parameter Groups by Dynamic Feedback:**

According to Lemma 1, for $g \in \mathcal{G}_{\text{tail}}$, we have:

$$\delta S_g \propto \frac{1}{N_k} \cdot \delta A_k.$$

Since $k$ is a tail class with small $N_k$, therefore:

$$\delta S_g \text{ is relatively large.}$$

Consequently, the importance scores $S_g$ of tail class parameter groups receive significant enhancement after each pruning round.

2. **Probability Calculation for Parameter Group Pruning:**

Pruning decisions are based on $S_g > S_{(\theta_t)}^{(t)}$, i.e.:

$$\mathbb{P}\left(S_g^{(t)} > S_{(\theta_t)}^{(t)}\right).$$

For $g \in \mathcal{G}_{\text{tail}}$, due to significant increase in $S_g$, the probability of being pruned decreases substantially.

3. **Comparison of Retention Probabilities between Two Pruning Algorithms:** - **Standard Pruning Algorithm** $\mathcal{P}$:

$$\mathbb{P}\left(\theta_g^{(t)} \neq \mathbf{0} \mid \mathcal{P}\right) = \mathbb{P}\left(S_g^{(t)} > S_{(\theta_t)}^{(t)} \mid \mathcal{P}\right).$$

Since $\mathcal{P}$ does not consider class accuracy changes, all parameter groups have similar retention probabilities. - **Dynamic Feedback Pruning Algorithm** $\mathcal{P}_{\text{LTAP}}$:

$$\mathbb{P}\left(S_g^{(t)} > S_{(\theta_t)}^{(t)} \mid \mathcal{P}_{\text{LTAP}}\right) \gg \mathbb{P}\left(S_g^{(t)} > S_{(\theta_t)}^{(t)} \mid \mathcal{P}\right).$$

4. **Cumulative Effect After Multiple Iterations:**

After $T$ rounds of pruning, for $g \in \mathcal{G}_{\text{tail}}$, we have:

$$\mathbb{P}\left(\theta_g^{(T)} \neq \mathbf{0} \mid \mathcal{P}_{\text{LTAP}}\right) = \prod_{t=1}^{T} \mathbb{P}\left(S_g^{(t)} > S_{(\theta_t)}^{(t)} \mid \mathcal{P}_{\text{LTAP}}\right).$$

$$\mathbb{P}\left(\theta_g^{(T)} \neq \mathbf{0} \mid \mathcal{P}\right) = \prod_{t=1}^{T} \mathbb{P}\left(S_g^{(t)} > S_{(\theta_t)}^{(t)} \mid \mathcal{P}\right).$$

Since for all $t$:

$$\mathbb{P}\left(S_g^{(t)} > S_{(\theta_t)}^{(t)} \mid \mathcal{P}_{\text{LTAP}}\right) \gg \mathbb{P}\left(S_g^{(t)} > S_{(\theta_t)}^{(t)} \mid \mathcal{P}\right),$$

therefore:

$$\mathbb{P}\left(\theta_g^{(T)} \neq \mathbf{0} \mid \mathcal{P}_{\text{LTAP}}\right) \gg \mathbb{P}\left(\theta_g^{(T)} \neq \mathbf{0} \mid \mathcal{P}\right).$$

Thus, the dynamic feedback pruning algorithm $\mathcal{P}_{\text{LTAP}}$ significantly increases the retention probability of parameter groups $\theta_g$ associated with tail classes.

# E. Pseudocode

---

0: **Input.** Pretraining variable $x$, learning rate $\beta$, termination tolerance $\mathcal{Z}$, preset pruning ratio $\gamma_p$, sample steps $T$, penalty $\lambda$, and prunable variable partition $\mathcal{G}$, class weight vector $w$.

0: Warm up $\mathcal{B}$ and compute importance scores.

0: Initialize $\mathcal{S}$ to store importance scores for each $g \in \mathcal{G}$.

0: Initialize violating group set $\mathcal{V}$

$$\mathcal{V} \leftarrow \{g : g \in \mathcal{G} \text{ with bottom-K importance scores}\}.$$

0: Initialize historical set $\mathcal{H} \leftarrow \mathcal{V}$.

0: **while** $|\mathcal{V}| \leq \mathcal{Z}$ **do**

0:     Initialize trial violating group set $\widehat{\mathcal{V}} \leftarrow \emptyset$.

0:     Initialize $\beta^0 \leftarrow \beta$, $\lambda^0 \leftarrow \lambda$, and $x^0 \leftarrow x$.

0:     **for** $t = 0, 1, \cdots, T-1$ **do**

0:         Compute gradient of $f$ over $x^{(t)}$ as $f(x^{(t)})$.

0:         Compute trial $\tilde{x}^{(t+1)} \leftarrow x^{(t)} - \beta^{(t)} f(x^{(t)})$.

0:         Penalize variables in the violating set.

$$[x^{(t+1)}]_{\mathcal{V}} \leftarrow [\tilde{x}^{(t+1)}]_{\mathcal{V}} - \lambda_t [x^{(t)}]_{\mathcal{V}}$$

0:         Compute the accuracy $A_c^{t+1}$ of the $x^{(t+1)}$ on each class.

0:         Update the importance criteria weight matrix.

$$\mathcal{D}^{(t+1)} \leftarrow (A_c^{t+1}, A_c^t, \mathcal{D}_t, w)$$

0:         Compute importance scores of $\mathcal{G}$ and collect into $\mathcal{S}$.

$$\mathcal{S} \leftarrow \mathcal{G} \leftarrow ([x^{(t+1)}]_{\mathcal{V}}, \mathcal{D}^{(t+1)})$$

0:         Update trial set $\widehat{\mathcal{V}}$ if new violating groups appear.

$$\widehat{\mathcal{V}} \leftarrow \widehat{\mathcal{V}} \cup \{g : g \in \mathcal{G} \text{ with bottom-K scores}\}/\mathcal{V}$$

0:         Update penalty $\lambda^{(t)}$ and learning rate $\beta^{(t)}$.

0:     **end for**

0:     Update violating set $\mathcal{V} \leftarrow \widehat{\mathcal{V}}/\mathcal{H}$.

0:     Update historical set $\mathcal{H} \leftarrow \mathcal{H} \bigcup \mathcal{V}$.

0: **end while**

0: Set redundant set $\mathcal{G}_R$ upon importance score collection $\mathcal{S}$.

$$\mathcal{G}_R \leftarrow \{g : g \text{ with bottom-K scores in } \mathcal{S}\}$$

0: **Return.** Identified redundant group set $\mathcal{G}_R$ and important group set $\mathcal{G}_I$ as $\mathcal{G}/\mathcal{G}_R$. =0

---

