# OpenReview forum: "Balancing Model Efficiency and Performance: Adaptive Pruner for Long-tailed Data"
_ICML.cc/2025/Conference — ICML 2025 poster_

### Official Review · Reviewer_A5yk · 2025-02-26

**Overall Recommendation:** 5

**Summary:**

This paper introduces Long-Tailed Adaptive Pruner (LTAP), a novel pruning strategy designed to enhance neural network efficiency while preserving performance on long-tailed datasets. LTAP addresses this challenge by incorporating multi-dimensional importance scoring and a dynamic weight adjustment mechanism, ensuring that essential parameters for tail classes are retained. The method employs progressive multi-stage pruning, gradually removing redundant parameters. Extensive experiments on multiple benchmark datasets demonstrate LTAP's effectiveness.

**Claims And Evidence:**

Yes.

**Essential References Not Discussed:**

Several relevant works of long-tailed learning have not been cited or discussed, including:

- Parametric Contrastive Learning, ICCV 2021
- Label-Imbalanced and Group-Sensitive Classification under Overparameterization, NeurIPS 2021
- Self-Supervised Aggregation of Diverse Experts for Test-Agnostic Long-Tailed Recognition, NeurIPS 2022
- Long-Tailed Recognition via Weight Balancing, CVPR 2022
- A Unified Generalization Analysis of Re-Weighting and Logit-Adjustment for Imbalanced Learning, NeurIPS 2023
- Balanced Product of Calibrated Experts for Long-Tailed Recognition, CVPR 2023
- Harnessing Hierarchical Label Distribution Variations in Test Agnostic Long-tail Recognition, ICML 2024

**Experimental Designs Or Analyses:**

The authors do not present the final criteria weight matrix $D$ in the experiments. It would be helpful to include it to validate the effectiveness of the weight adjustment mechanism.

**Methods And Evaluation Criteria:**

- In Section 2.3, the authors state that the updating rule strengthens the criteria that lead to improved class performance. However, in Eq. (6), it appears that if the accuracy of class $c$ improves, the weights for all criteria of class $c$ are increased by the same value $\beta$. This suggests that the method does not actually select the most effective criteria for class $c$. More explanation is needed to clarify this point.

- The CIFAR-LT-100 dataset with an imbalance ratio of 10 is widely used. Including this dataset could be better but is not strictly necessary.

**Other Comments Or Suggestions:**

- There are some typos. In the Implementation Details section, there appears to be a formatting issue with the superscripts in the learning rate and weight decay values.

**Other Strengths And Weaknesses:**

**Strengths**

- Addressing model pruning in the presence of long-tailed data distributions is both meaningful and highly relevant to real-world scenarios.

- The proposed method achieves a relatively significant performance improvement.

**Weaknesses**

- The novelty of the method is somewhat limited. It primarily focuses on combining different criteria with adaptive weights. However, as mentioned in the *Method and Evaluation Criteria* part, the weight adjustment mechanism does not seem effective in selecting the most effective criteria. This aspect may require further explanation.
- Some notations are not properly defined. For example, the definition of $n_g$ in Eq. (3) and $\beta$ in Eq. (6) is missing.

**Questions For Authors:**

Please see above.

**Relation To Broader Scientific Literature:**

The proposed method contributes to the broader scientific literature by tackling the challenges of model pruning in the presence of long-tailed data distributions.

**Theoretical Claims:**

Yes.

---

> ### Author Rebuttal · Authors · 2025-04-01
>
> **Thank you for your recognition of our work. We have carefully understood the relevant weaknesses you mentioned and have made the following efforts:**
>
> - **About broader references**
>
> We have incorporated the studies you mentioned into our discussion and references as follows:
>
> Recent advances in long-tailed learning address class imbalance through diverse strategies:
>
> 1. **Theoretical & Unified Frameworks**: [Wang et al., 2023] proposed a data-dependent contraction technique to unify re-weighting and logit adjustment.
> 2. **Hierarchical Label Variation**: [Yang et al., 2023] introduced **DirMixE**, leveraging Dirichlet meta-distributions to model global-local test-time variations, with variance regularization for stable generalization.
> 3. **Ensemble Calibration**: [Aimar et al., 2023] formulated **BalPoE**, a Fisher-consistent ensemble combining logit-adjusted experts calibrated via mixup.
> 4. **Weight Regularization**: [Alshammari et al., 2023] demonstrated that simple weight decay and MaxNorm constraints outperform complex rebalancing methods.
> 5. **Contrastive Rebalancing**: [Cui et al., 2023] proposed **PaCo**, integrating parametric class centers into contrastive learning to mitigate gradient bias toward head classes.
> 6. **Margin Adjustment Theory**: [Kini et al., 2023] analyzed **VS-loss**, combining additive/multiplicative logit adjustments to optimize margins in overparameterized regimes.
> 7. **Test-Agnostic Aggregation**: [Zhang et al., 2022] developed **SADE**, a self-supervised expert aggregation framework for unknown test distributions.
>
> These works collectively advance the field by bridging theory-practice gaps [1,6], enhancing model flexibility [2,7], and simplifying regularization [4], while addressing both label and group imbalances [3,5,6].
>
> - **About the supplementary experiment on CIFAR-100-LT (IR=10)**
>
> According to your suggestion, we supplemented the comparison experiment with IR=10, and the results are shown in the table below.
>
> |Method|F(%)↓|Head↑|Medium↑|Tail↑|All↑|C(%)↑|C/F↑|
> |-|-|-|-|-|-|-|-|
> |BS|100.0|64.9|55.5|0.0|61.5|100.0|1.0|
> |BS + ATO|84.7|45.1|37.2|0.0|41.1|66.8|0.8|
> |BS + RReg|52.1|46.2|38.7|0.0|42.5|69.1|1.3|
> |BS + ours|22.6|56.4|47.3|0.0|53.8|87.4|3.8|
> |LDAM-DRW|100.0|60.8|43.3|0.0|55.4|100.0|1.0|
> |LDAM-DRW + ATO|84.7|41.5|26.2|0.0|37.3|67.3|0.8|
> |LDAM-DRW + RReg|52.1|39.8|25.1|0.0|35.2|63.5|1.2|
> |LDAM-DRW + ours|22.6|51.9|35.7|0.0|47.1|85.0|3.7|
> |DBLP|100.0|65.3|43.4|0.0|58.7|100.0|1.0|
> |DBLP + ATO|84.7|49.5|28.3|0.0|42.3|72.0|0.9|
> |DBLP + RReg|52.1|48.3|27.2|0.0|41.1|70.1|1.3|
> |DBLP + ours|22.6|57.3|30.0|0.0|49.2|83.8|3.7|
>
>
> - **About criteria weight matrix**
>
> To facilitate your understanding, we have supplemented the images of the weight matrix $D$ in the last training process, where the horizontal axis represents the index of the class, different color blocks represent different pruning weight calculation methods, and the length of different color blocks represents the numerical value of different pruning weights in the weight matrix $D$.
>
> [https://anonymous.4open.science/r/AEFCDAISJ/D_matrix_pic.png](https://anonymous.4open.science/r/AEFCDAISJ/D_matrix_pic.png)
>
> - **About weight adjustment mechanism**
>
> The observations you make are very sharp. In a single step of equation (6), for the improved category $c$, the weights of all relevant criteria indeed increase by the same value $\beta$. We understand that this may raise questions about how the mechanism actually chooses the 'most effective criteria', and we provide the following clarifications in this regard:
>
> (i) Equation (6) is not designed to directly identify a single 'most efficient criterion', but rather to implement a more nuanced and multi-dimensional dynamic balancing mechanism.
>
> (ii) Among other things, the reasons for this design rather than directly identifying the 'best criteria' are:
>
> 1. In practice, multiple criteria often work together to achieve the best results, rather than a single criterion being dominant.
> 2. As reviewer LjiE said, the core goal of this paper lies in introducing distribution-aware capabilities for pruning strategies. The above standard is the carrier of this distribution-aware capability to effective pruning. In future work, we plan to explore mechanisms for more precise and interpretable distribution-aware parameter pruning.
> - **About novelty**
>
> As stated in the previous point, our goal is to introduce distributional awareness to pruning, and to this end, LTAP redefines the pruning problem in the long-tailed scenario from 'uniform compression' to 'differentiated parameter assignment', a conceptual shift that is significantly innovative in its own right. **This method of directly correlating distribution properties with parameter importance opens a new path for the application of neural network compression on imbalanced data.**
>
> - **About notation and spelling errors**
>
> We appreciate your comments and will fully revise them in a subsequent version.

---

> > ### Comment · Reviewer_A5yk · 2025-04-03
> >
> > I have reviewed the author's response. I am satisfied with the author's efforts. The supplementary experiments fully demonstrate the necessity and rationality of the proposed method. Additionally, the author's additional explanations regarding novelty and motivation are clear to me. I also agree with the recognition of the theoretical and experimental contributions of this paper by other reviewers, as well as their views on its significance in the long-tail domain. I have decided to raise my score.

---

### Official Review · Reviewer_T1Bx · 2025-03-09

**Overall Recommendation:** 3

**Summary:**

This paper introduces an adaptive pruning method called LTAP to address the challenge of handling long-tailed distribution data. The authors propose a multi-dimensional importance scoring criterion and design a dynamic weight adjustment mechanism to adaptively determine the pruning priority of parameters for different classes. Experimental results on various benchmark datasets, such as CIFAR-100-LT and ImageNet-LT, demonstrate improvements in both computational efficiency and classification accuracy for tail classes.

**update after rebuttal**

Thank you for your rebuttal. I will keep my score unchanged and remain positive about this paper.

**Claims And Evidence:**

Please refer to Strengths And Weaknesses.

**Essential References Not Discussed:**

Please refer to Strengths And Weaknesses.

**Experimental Designs Or Analyses:**

Please refer to Strengths And Weaknesses.

**Methods And Evaluation Criteria:**

Please refer to Strengths And Weaknesses.

**Other Comments Or Suggestions:**

Please refer to Strengths And Weaknesses.

**Other Strengths And Weaknesses:**

**Strengths**
1. The concept of adaptive pruning to balance model efficiency and performance on long-tailed data is intriguing and has the potential to attract interest from the research community.
2. The author provides theoretical analysis to strengthen the persuasiveness of the proposed method.
3. The overall organization of the paper is well-structured.

**Weaknesses**
1. Personally, I believe that instead of presenting numerous theorems and textual explanations in Section 3, incorporating visual elements would be more effective. This could help readers gain deeper insights into the methodology more easily.
2. A minor issue: Some symbols used in the formulas are not well explained, sometimes I feel confused when reviewing this paper. I suggest that the author improve the writing and enhance the clarity of the paper’s content.

**Questions For Authors:**

Please refer to Strengths And Weaknesses.

**Relation To Broader Scientific Literature:**

Please refer to Strengths And Weaknesses.

**Theoretical Claims:**

Please refer to Strengths And Weaknesses.

---

> ### Author Rebuttal · Authors · 2025-04-01
>
> **Thank you for your recognition of our work. We have carefully understood the relevant weaknesses you mentioned and have made the following efforts:**
>
> - **About presentation form**
>
> Thank you for your valuable suggestions. We will supplement the theorems and textual explanations in Section III with some visual instructions to help the reader understand. We will illustrate (i) why pruning in the long-tail scenario requires special protection, and (ii) the core idea of distribution-aware pruning through a schematic illustration.
>
> - **About symbolic expression**
>
> Thank you for your advice. We carefully proofread the notations used in this paper and improved some inappropriate descriptions.
>
> **Thank you again for your comments and we will continue to work on improving the quality of the manuscript！**

---

### Official Review · Reviewer_LjiE · 2025-03-16

**Overall Recommendation:** 4

**Summary:**

This paper introduces Long-Tailed Adaptive Pruner (LTAP), a model pruning framework designed for long-tailed class distributions. LTAP integrates multi-criteria importance scoring and a dynamic LT-Vote mechanism to prioritize preserving parameters crucial for tail classes. It employs multi-stage pruning, gradually refining the model while maintaining performance. Theoretical analysis supports the claim that tail classes require more capacity, and experiments on CIFAR-100-LT, ImageNet-LT, and iNaturalist 2018 demonstrate that LTAP outperforms traditional pruning and long-tail learning methods, improving tail-class accuracy while reducing model size by 70%. LTAP provides a new approach to balancing efficiency and fairness in imbalanced learning.

**Claims And Evidence:**

The paper makes three major claims: (a) Tail classes require more model capacity than head classes, (b) Dynamically adjusting pruning criteria improves tail-class retention, and (c) LTAP achieves better accuracy-efficiency trade-offs than existing methods.
Overall, the claims are well-supported by theoretical proofs and extensive experiments across multiple datasets. The results consistently show LTAP’s advantage in balancing efficiency and tail-class performance.

**Essential References Not Discussed:**

While the paper includes strong baselines such as LDAM-DRW and DBLP, there exist many other long-tailed learning approaches that could have further strengthened the evaluation. Methods like Focal Loss, BBN, and Logit Adjustment are widely used in long-tailed classification but were not explicitly included in the comparisons. Additionally, some recent state-of-the-art long-tailed learning strategies, such as MiSLAS and RIDE, which utilize representation learning and multi-expert models, could have been relevant baselines. While this omission does not undermine the paper’s key contributions, including a broader range of baselines would have further solidified LTAP’s effectiveness by demonstrating its adaptability across different long-tail learning paradigms.

**Experimental Designs Or Analyses:**

The experimental design is rigorous and well-structured, validating LTAP across CIFAR-100-LT, ImageNet-LT, and iNaturalist 2018, covering both controlled and real-world imbalance scenarios. The method consistently improves tail-class accuracy while reducing FLOPs, with results showing higher accuracy-per-FLOP (C/F) than competing methods. The evaluation metrics include head/medium/tail accuracy breakdowns, highlighting tail-class improvements.

**Methods And Evaluation Criteria:**

The methodology is well-designed, effectively addressing pruning bias in imbalanced data while maintaining efficiency. The evaluation is comprehensive, though hyperparameter robustness (e.g., pruning schedule, weight updates) remains an open question.

**Other Comments Or Suggestions:**

The text in Figure 2 is too small, making it difficult to read.

**Other Strengths And Weaknesses:**

Please refer to the above response.

**Questions For Authors:**

How does an LTAP-pruned model (30% parameters retained) compare to a manually designed smaller model (with ~30% of the original parameters) trained with long-tail techniques? Does LTAP find a more effective sub-network than simply training a smaller model from scratch?

**Relation To Broader Scientific Literature:**

This paper bridges long-tailed learning and model pruning, two traditionally separate research areas, by introducing adaptive pruning tailored for class imbalance. Unlike prior long-tail learning methods that focus on reweighting, resampling, or architectural modifications (e.g., expert models, transfer learning, logit adjustments), LTAP optimizes model structure dynamically to preserve tail-class critical parameters, making it a novel contribution to long-tail research. Similarly, while pruning methods have primarily targeted efficiency and overall accuracy, LTAP introduces a distribution-aware pruning strategy that prioritizes fairness across head and tail classes.

**Theoretical Claims:**

The paper presents a strong theoretical foundation supporting LTAP’s approach, with multiple theorems validating its core ideas. These claims are supported by rigorous proofs in the appendix, which follow established generalization theory and sample complexity principles. While some assumptions may not hold perfectly in practice, the results provide a strong theoretical justification for LTAP's adaptive pruning strategy. The presence of formal guarantees enhances the paper’s credibility, making LTAP a well-grounded contribution to long-tailed learning and pruning research.

---

> ### Author Rebuttal · Authors · 2025-04-01
>
> **Thank you for your time, effort and recognition of our work. Your comments are very important for us to continue to improve this work. Based on your comments, we continue to make the following efforts**:
>
> - **About Pseudocode**
>
> This is our negligence. We have completed and corrected the pseudocode in the original paper.
>
> - **About the broader baselines**
>
> Thank you for your approval of the experimental section. Based on your comments, **we tried our best to supplement some baselines in the limited time, including Focal Loss, Logit Adjustment, and RIDE**. The following are the experimental results regarding supplemental baseline on the CIFAR-100-LT dataset. The experimental results show the superior performance of LTAP.
>
> |Method|F(%)↓|IR=10 Head↑|IR=10 Medium↑|IR=10 Tail↑|IR=10 All↑|IR=10 C(%)↑|IR=10 C/F↑|IR=50 Head↑|IR=50 Medium↑|IR=50 Tail↑|IR=50 All↑|IR=50 C(%)↑|IR=50 C/F↑|IR=100 Head↑|IR=100 Medium↑|IR=100 Tail↑|IR=100 All↑|IR=100 C(%)↑|IR=100 C/F↑|
> |-|-|-|-|-|-|-|-|-|-|-|-|-|-|-|-|-|-|-|-|
> |Focal Loss|100.0|65.1|43.4|0.0|58.8|100.0|**1.0**|67.2|38.6|13.8|46.2|100.0|**1.0**|67.8|39.3|8.0|40.2|100.0|**1.0**|
> |Focal Loss + ours|22.0|57.1|32.3|0.0|49.7|84.5|**3.8**|61.9|29.6|5.4|38.7|83.7|**3.8**|63.4|35.9|4.8|36.4|90.5|**4.1**|
> |logit adjust|100.0|57.5|63.2|0.0|59.4|100.0|**1.0**|59.3|46.7|43.2|51.4|100.0|**1.0**|62.4|47.1|27.9|46.9|100.0|**1.0**|
> |logit adjust + ours|22.0|45.2|58.3|0.0|49.4|83.1|**3.7**|50.7|43.7|36.4|45.5|88.5|**4.0**|55.1|45.5|23.0|42.5|90.6|**4.1**|
> |RIDE|100.0|70.5|42.0|0.0|61.6|100.0|**1.0**|68.5|48.8|44.0|51.1|100.0|**1.0**|68.1|49.2|23.9|48.0|100.0|**1.0**|
> |RIDE + ours|22.0|62.8|32.5|0.0|53.4|86.6|**3.9**|62.8|40.1|32.0|43.9|85.9|**3.9**|62.0|46.8|18.2|45.3|90.6|**4.1**|
>
>
> - **About Figure 2**
>
> Thanks for your suggestion, we have adjusted the font of Figure 2 in the original paper to improve readability.
>
> - **About Questions**
>
> Indeed, the comparison with smaller models is an interesting point, which is fundamental to pruning studies but has unique significance in long-tailed scenarios.
>
> First, as visualizations in Figures 2 to 6 demonstrate, **LTAP pruned parameters in a non-uniform, category-aware manner, which is difficult to achieve with manual architecture design**. By simultaneously considering multiple importance metrics and adjusting their weights based on accuracy feedback, LTAP may identify subtle parameter interactions that might have been missed in simple architectural reductions. This is fundamentally different from simply reducing the model size uniformly.
>
> In addition, we try our best to organized a contrast experiment: the performance comparison between the manually designed small model and the original standard model, where the flops of the small model is 30% of the original standard model. If needed, you may kindly compare this with Table 1 in the original paper.
>
> |Method|F(%)↓|IR=10 Head↑|IR=10 Medium↑|IR=10 Tail↑|IR=10 All↑|IR=10 C(%)↑|IR=10 C/F↑|IR=50 Head↑|IR=50 Medium↑|IR=50 Tail↑|IR=50 All↑|IR=50 C(%)↑|IR=50 C/F↑|IR=100 Head↑|IR=100 Medium↑|IR=100 Tail↑|IR=100 All↑|IR=100 C(%)↑|IR=100 C/F↑|
> |-|-|-|-|-|-|-|-|-|-|-|-|-|-|-|-|-|-|-|-|
> |BS|100.0|64.9|55.5|0.0|61.5|100.0|1.0|62.3|46.1|37.0|51.2|100.0|1.0|62.6|48.5|27.0|47.2|100.0|1.0|
> |BS|30.0|54.8|45.1|0.0|52.1|84.7|2.8|53.3|40.8|30.0|44.1|86.1|2.8|55.1|41.6|21.0|40.4|85.5|2.8|
> |LDAM-DRW|100.0  |60.8  |43.3|0.0|55.4|100.0|1.0|64.5|43.0|26.4|49.1|100.0|1.0|65.1|48.1|20.1|45.8|100.0|1.0|
> |LDAM-DRW|30.0|51.0|30.1|0.0|45.0|81.2|2.7|54.8|34.1|20.4|40.2|81.8|2.7|56.0|36.0|14.6|36.9|80.3|2.6|
> |DBLP|100.0|65.3|43.4|0.0|58.7|100.0|1.0|61.2|46.5|32.3|50.2|100.0|1.0|61.4|46.9|23.6|45.3|100.0|1.0|
> |DBLP|30.0|57.9|30.0|0.0|49.5|84.3|2.8|62.8|29.3|5.5|38.9|77.4|2.5|63.7|32.1|3.3|34.6|76.3|2.5|

---

### Official Review · Reviewer_xkjm · 2025-03-17

**Overall Recommendation:** 3

**Summary:**

This paper introduces ​LTAP (Long-Tailed Adaptive Pruner), a pruning strategy tailored for long-tailed data distributions. LTAP addresses the challenge of class imbalance by dynamically adjusting pruning priorities through a multi-criteria importance evaluation framework.

**Claims And Evidence:**

Formal proofs establish that tail classes inherently demand higher overparameterization, justifying LTAP’s tail-biased parameter retention strategy.

**Essential References Not Discussed:**

NA.

**Experimental Designs Or Analyses:**

LTAP achieves state-of-the-art efficiency-accuracy trade-offs across multiple architectures (ResNet-32/50) and datasets.

**Methods And Evaluation Criteria:**

Extensive experiments on CIFAR-100-LT, ImageNet-LT, and iNaturalist 2018 demonstrate LTAP’s superiority over baseline methods.

**Other Comments Or Suggestions:**

NA.

**Other Strengths And Weaknesses:**

Strength:

LTAP is the first pruning framework explicitly designed for long-tailed data. The LT-Vote mechanism effectively mitigates pruning bias toward head classes by dynamically reweighting criteria (magnitude, gradient alignment, Taylor impact) based on per-class validation accuracy. This innovation directly addresses the core challenge of class imbalance in pruning.

Overall, this work makes a contribution to long-tailed learning by bridging pruning and class imbalance mitigation. The LTAP framework is both theoretically grounded and empirically robust, offering a practical solution for efficient model deployment.

**Questions For Authors:**

NA.

**Relation To Broader Scientific Literature:**

NA.

**Theoretical Claims:**

The paper provides theoretical analysis (e.g., Theorem 1–4) to justify why tail classes require higher parameter protection.

---

> ### Author Rebuttal · Authors · 2025-04-01
>
> Thank you for your time and effort. We are encouraged by your high appreciation of the novelty and contribution of our paper. We will continue to work on exploring how to better solve long-tailed problems in real-world scenarios. If you have any further questions, we will address them at any time. Thanks again!

---

### Decision · Program_Chairs · 2025-05-01

**Decision:**

Accept (poster)

**Comment:**

After review, the paper received four positive evaluations. Following the authors' rebuttal, two reviewers increased their ratings, while the other two maintained their original scores. All reviewers acknowledged the paper's contributions and expressed satisfaction with its technical merits.

The AC concurs with the reviewers' assessments and recommends acceptance.